

# Use of nonlinear principal components of CHIRPS precipitation data and ocean-atmospheric variables for streamflow forecasting in an area of scarce data. Case study, Tocaría river basin - Orinoquia Colombiana

Sarria-Ospina, Jhon Derly[1]; Ocampo-Marulanda, Camilo[2,3]; Ceron-Aramburo, Lina Maria[1]; Canchala, Teresita[4]; Ferreira, Tiago Alessandro[2]

[1]Research Group TERRANARE, Faculty of Natural Sciences and Engineering, Fundación Universitaria de San Gil, Yopal, 850001, Colombia
[2]Programa de Pós-graduação em Biometria e Estatística Aplicada, Departamento de Estatística e Informática, Universidade Federal Rural de Pernambuco, Recife, 52171-900, Brasil
[3]Water Resources, Engineering and Soil Research Group (IREHISA), School of Natural Resources and Environmental Engineering, Universidad del Valle, 760032, Cali, Colombia
[4]Environmental Engineering Program, Faculty of Engineering, Universidad Mariana, Pasto, 520002, Colombia

*Correspondence to:* Sarria-Ospina, Jhon (jsarria.sgryopal@unisangil.edu.co)

**Abstract.** Accurate streamflow forecasting is critical for mitigating the impacts of hydrological extremes and guiding sustainable water resource management, particularly in poorly gauged tropical catchments. This study presents a hybrid forecasting framework that integrates Neural Network Seasonal Autoregressive Integrated Moving Average using exogenous variables (NN-SARIMAX) models with nonlinear principal components (NLPCs) derived from CHIRPS precipitation data, and large-scale ocean–atmosphere indices (macroclimatic variables, MVs). Four monthly models were developed and tested for the Tocaría River basin in the Colombian Orinoquía region: (1) a baseline SARIMA $(4,0,4)$ $(0,0,3)_{12}$ model; (2) SARIMAX with exogenous MVs; (3) NN-SARIMAX with NLPCs; and (4) a hybrid NN-SARIMAX combining both MVs and NLPCs. The hybrid model achieved the best performance with an $R^2$ of 0.78 during the validation period. These results underscore the effectiveness of integrating local precipitation variability and large-scale climatic drivers to enhance forecast accuracy under data-scarce conditions. The proposed methodology offers a transferable approach for operational forecasting in ungauged or sparsely monitored basins, contributing to early warning systems, drought preparedness, and adaptive water governance in vulnerable tropical regions.

**Keywords:** monthly forecasting, macroclimatic variables, NLPC, SARIMA, exogenous variables

# 1 INTRODUCTION

Sustainable management of water resources in tropical basins critically depends on the ability to accurately anticipate streamflow dynamics amid complex hydroclimatic variability. In regions like the Colombian Orinoquía, the interplay of large-scale ocean–atmosphere phenomena; especially the El Niño–Southern Oscillation (ENSO); introduces pronounced



nonstationarity and nonlinearity, challenging traditional time series forecasting methods (Pasquini et al., 2007; Chiew and McMahon, 2002; Yap and Musa, 2023). Streamflow integrates precipitation, evapotranspiration, infiltration, and climatic anomalies across multiple spatial and temporal scales, yet sparse hydrometeorological monitoring limits the characterization

of these processes (Tootle et al., 2008; Yao et al., 2020).

Conventional statistical models such as Autoregressive Integrated Moving Average (ARIMA) and its seasonal extension (SARIMA) have been widely applied for streamflow prediction due to their capacity to model temporal dependencies and seasonality under stationarity assumptions (Valipour, 2015; Sirisha et al., 2022). However, their performance diminishes in capturing abrupt, nonlinear fluctuations characteristic of tropical catchments influenced by exogenous climate drivers (Moeeni

and Bonakdari, 2017). Consequently, hybrid approaches that couple SARIMA with Artificial Neural Networks (ANNs) have emerged, leveraging ANNs' nonlinear learning capacity and robustness to data scarcity to improve forecasting skill (Rafael et al., 2022; Niu and Feng, 2021; Costa et al., 2023).

In Colombia, ANN-based models incorporating macroclimatic indices; such as sea surface temperature anomalies and ENSO-related indices; have demonstrated substantial improvements in streamflow forecasting accuracy for Andean and Pacific basins

(Mesa et al., 2001; Poveda et al., 2002; Velásquez et al., 2010; Cárdenas et al., 2022). Nevertheless, similar studies remain scarce for the Orinoquía region, despite its hydrological sensitivity to both Pacific and Atlantic climatic influences (Builes-Jaramillo et al., 2022). The Tocaría River basin (Orinoquía region), exemplifies this knowledge gap; it is characterized by limited hydrometeorological instrumentation and experiences seasonal extremes of drought and flooding that impact local agriculture and ecosystems (Corporinoquia and Corpoboyacá, 2015a).

Building on previous work that successfully employed NLPCA combined with satellite-derived CHIRPS precipitation data to impute missing monthly streamflow records in the Tocaría basin; demonstrating high accuracy across stations with substantial data gaps (Ocampo-Marulanda et al., 2025); this study advances the development of a parsimonious hybrid forecasting framework. The proposed model integrates a SARIMA component with feedforward ANNs and incorporates NLPC derived from CHIRPS data alongside macroclimatic variables representing large-scale atmospheric–oceanic drivers named NN-

SARIMAX. This nonlinear dimensionality reduction enhances model generalization in low-data contexts, effectively capturing the concurrent influences of Pacific and Atlantic climate variability.

The research aims to provide an accurate monthly streamflow forecasting tool tailored to the Tocaría River's hydrological regime, supporting adaptive water resource management in a data-limited and climate-vulnerable tropical basin. By addressing key gaps in scientific knowledge and practical decision-making tools, the framework offers scalable potential for application

in other similarly under-monitored tropical catchments subject to complex climatic forcing.



## 2 METHODOLOGY

This section outlines the methodological framework developed for monthly streamflow forecasting in poorly instrumented basins, using the Tocaría River (Colombian Orinoquia) as a case study. The approach integrates seasonal autoregressive modeling with exogenous predictors, nonlinear dimensionality reduction, and multivariate correlation analysis.

As illustrated in Fig 1, the workflow combines three primary data sources—observed streamflow, satellite-derived precipitation (CHIRPS), and ocean–atmosphere macroclimatic variables (MVs)—through preprocessing, imputation, NLPCA-based reduction, and statistical evaluation. Four forecasting models were independently constructed and trained, and subsequently compared using RMSE, R², Akaike Information Criterion (AIC), Bayesian Information Criterion (BIC), and residual diagnostics.


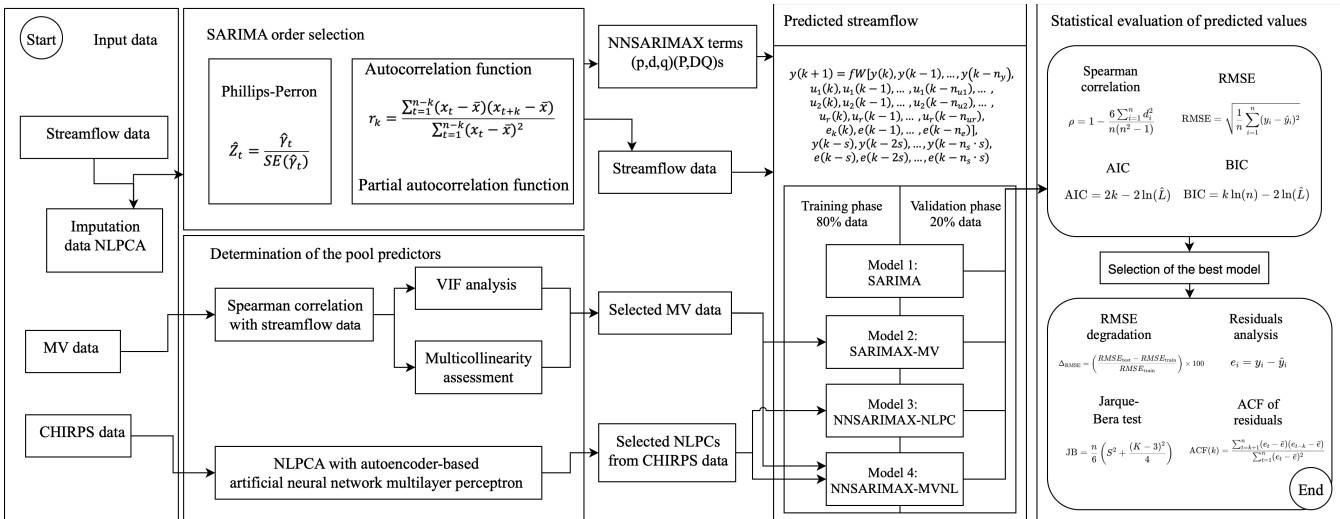

**Figure 1: Schematic representation of the methodological workflow for streamflow forecasting under data-scarce conditions.**

### 2.1 Study area

The Tocaría River originates at 3,200 m above sea level in the Guevarrica hill and drains a watershed of 2,223 km² on the
eastern flank of the eastern cordillera. The river extends over 127.5 km with an average longitudinal slope of 5%. The basin is part of the Cravo Sur system, a tributary to the Meta River within the Orinoco macro-basin (Corporinoquia, 2010; Corporinoquia and Corpoboyacá, 2015b) (Fig. 2).

The basin experiences a monomodal rainfall regime with a distinct wet season from April to November and a dry season from December to March. Although 90% of the basin lies within the Andean region, its discharge contributes to the Orinoquía
domain. Mean annual precipitation is approximately 2,031 mm, with peak streamflows during the wet season and minimum flows in February (Ruíz-Ochoa et al., 2022; Urrea et al., 2016).



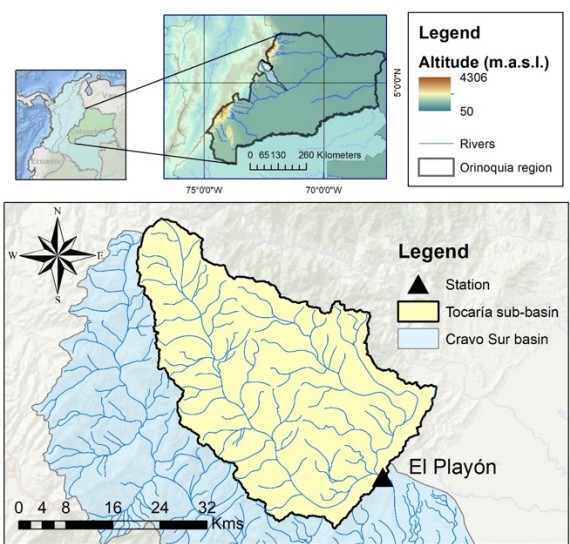

**Figure 2: Geographic location of the Tocaría River basin and El Playón hydrometric station. Digital elevation model, political boundaries, watershed boundaries, and rivers extracted from GEOPORTAL IDEAM: https://archivo.ideam.gov.co/geoportal**

## 2.2 Data

Three main datasets were used for model development: observed streamflow, satellite-based precipitation, and large-scale macroclimatic indices.

Streamflow data were obtained from the El Playón station, operated by Colombia's national hydrometeorological institute (IDEAM), covering 1983–2019 with approximately 5% missing values. Data were sourced from the DIHME geoportal (http://dhime.ideam.gov.co). Missing values were imputed using NLPCA, incorporating CHIRPS precipitation data as an exogenous input, following the approach by Ocampo-Marulanda et al. (2025).

Mean monthly discharge averaged 85 m³/s, with strong seasonal variability: February averaged 17 m³/s (dryest month), while July peaked at 172 m³/s (Fig. 3). These fluctuations highlight the necessity of modeling strategies that capture both hydroclimatic seasonality and interannual variability.





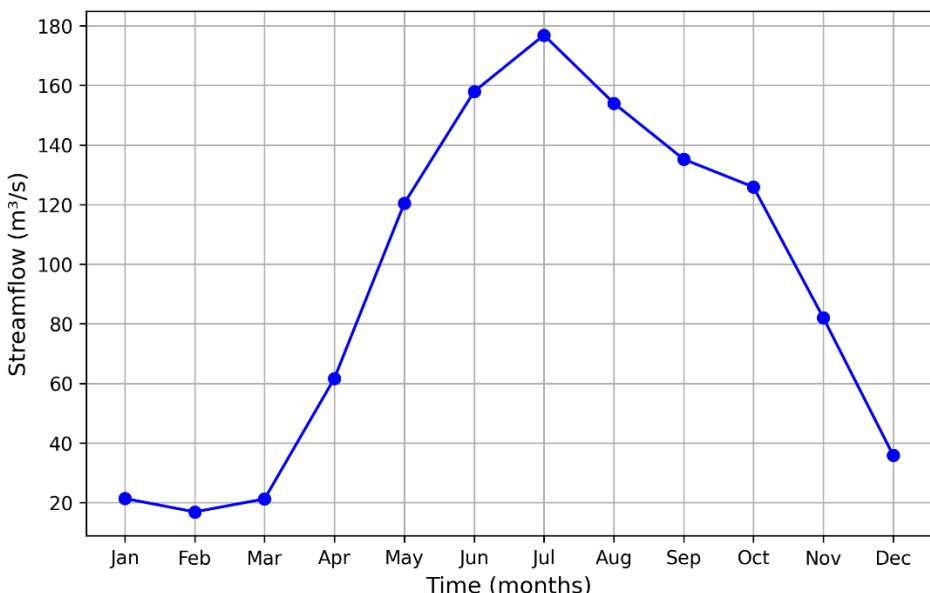

**Figure 3: Monthly streamflow climatology at the El Playón station, Tocaría River.**

Macroclimatic variables were retrieved from the National Oceanic and Atmospheric Administration (NOAA, https://psl.noaa.gov/gcos_wgsp/). Twenty-one indices representing sea surface temperature anomalies, atmospheric pressure, and wind patterns were selected based on prior demonstrated correlations with Colombian streamflow regimes (Poveda et al., 2011; Cerón et al., 2020; Canchala et al., 2020a). Complete descriptions of selected variables are available in Supplementary Table S1.

Precipitation data were sourced from the Climate Hazards Group InfraRed Precipitation with Station data (CHIRPS), which integrates satellite and in situ observations. CHIRPS provides global coverage at 0.05° spatial resolution from 1981 to present. Its performance in Colombia is well documented (Funk et al., 2014; Urrea et al., 2016; Ocampo-Marulanda et al., 2022). In this study, CHIRPS data were used to derive nonlinear principal components (NLPC) from 81 time series corresponding to 81 grid cells located within the Tocaría River basin.

**2.3 Data preprocessing**

This subsection details statistical preprocessing applied to streamflow data prior to model development, including variability characterization, trend detection, structural change identification, and stationarity assessment.

**2.3.1 Descriptive statistics and trend analysis**

Descriptive statistics calculated included mean ($\mu$), maximum, standard deviation ($\sigma$), and coefficient of variation (CV),

providing an initial quantification of streamflow magnitude and variability.





Structural changes were identified via Pettitt's test (Pettitt, 1979), revealing a significant breakpoint in the time series mean. Sen's slope estimator (Sen, 1968) quantified long-term trends, while the Mann-Kendall test (Mann, 1945) evaluated trend significance.

### 2.3.2 Stationarity assessment

Stationarity was assessed through additive time series decomposition into trend, seasonal, and residual components, facilitating visualization of nonstationary behavior.

The Phillips–Perron (PP) test evaluated the null hypothesis of a unit root, offering robustness to autocorrelation and heteroskedasticity in residuals. Non-rejection of the null indicated nonstationarity, guiding NN-SARIMAX model structure selection.

### 2.3.3 Autocorrelation and Partial Autocorrelation

Temporal dependencies were examined via autocorrelation function (ACF) and partial autocorrelation function (PACF) analyses (Chatfield, 1989). The ACF quantifies linear correlation at lag $k$, defined as:

$$r_k = \frac{\sum_{t=1}^{n-k}(x_t-\bar{x})(x_{t+k}-\bar{x})}{\sum_{t=1}^{n-k}(x_t-\bar{x})^2} \tag{1}$$

where $x_t$ is the series value at time $t$, $\bar{x}$ is the mean, and $n$ the number of observations.

PACF measures correlation at lag k controlling for intermediate lags, informing autoregressive and moving average term selection in NN-SARIMAX models (Box et al., 2015). ACF and PACF plots were examined for seasonality and persistence, using 95% confidence intervals to identify significant lags.

### 2.4 Selection of possible correlators

### 2.4.1 CHIRPS precipitation as a predictor

The hydrological relationship between precipitation and streamflow in the Cravo Sur basin has been previously demonstrated (Ocampo-Marulanda et al., 2025), supporting the use of CHIRPS precipitation data as potential predictors. In this study, 81 precipitation time series corresponding to CHIRPS grid cells within the Tocaría sub-basin were analyzed.

To prevent overfitting and reduce dimensionality, NLPCA was applied to extract two principal components capturing dominant nonlinear spatiotemporal precipitation patterns. Implemented via an autoencoder with bottleneck architecture, NLPCA
minimizes reconstruction error between inputs and outputs, capturing nonlinear relationships beyond traditional PCA capabilities (Scholz et al., 2007; Canchala et al., 2019). Validation of extracted components included reconstruction accuracy and latent space visualization, confirming their efficacy in summarizing precipitation variability.



### 2.4.2 Macroclimatic variables as large-scale predictors

MVs comprising sea surface temperature anomalies, sea-level pressure, and wind indices known to influence Colombian hydrology (Canchala et al., 2020a,b; Cerón et al., 2020; Poveda et al., 2001) were evaluated.

Spearman rank correlations between each MV and streamflow were calculated for lags up to 14 months to capture delayed effects. Variables exhibiting statistically significant correlations at lags ≥ 6 months were retained, reflecting hydroclimatic memory. Only predictors with demonstrated hydrological relevance and minimal intercorrelation were preserved to ensure model parsimony and stability.

### 2.4.3 Multicollinearity diagnosis and variable selection


To ensure stable and interpretable forecasting models, a two-step assessment of multicollinearity was performed. Initially, correlation matrices visualized as heatmaps identified groups of highly correlated predictors that could introduce redundancy and overfitting risks (Dormann et al., 2013). Subsequently, the Variance Inflation Factor (VIF) quantified the degree of linear dependence among predictors (Katrutska and Strijov, 2017). Predictors with VIF values exceeding the threshold of 10 were

flagged for potential exclusion.

However, exclusion was not automatic. Variables exhibiting high VIF but demonstrating strong and meaningful correlations with streamflow were retained to preserve model interpretability and relevance (Lavery et al., 2017). Conversely, predictors with both high VIF and weak correlation with the target variable were removed. This iterative process was coupled with evaluation of model performance across different predictor subsets, balancing parsimony, predictive accuracy, and explanatory

power. The final predictor set thus optimized robustness and transparency in streamflow forecasting.

### 2.5 Forecasting model: NN-SARIMAX architecture

The forecasting approach proposed herein is based on a hybrid NN-SARIMAX framework that integrates nonlinear temporal dynamics, exogenous climatic forcings, and seasonal variability within a unified predictive architecture. This model synergizes the statistical generalization capacity of SARIMA structures with the universal function approximation capabilities of neural

networks trained on NLPC extracted from high-dimensional hydroclimatic inputs.

The forecasted streamflow vector, denoted as $\hat{y}(k) \in R^m$, is modeled as a nonlinear function of multiple lagged sequences:

Streamflow autoregressive terms:

$$y(k), y(k-1), \dots, y(k-n_y) \tag{2}$$

Exogenous forcings (e.g., ocean-atmospheric indices):

$$u_1(k), u_1(k-1), \dots, u_1(k-n_{u1}), \dots; \dots; u_r(k), u_r(k-1), \dots, u_r(k-n_{ur}) \tag{3}$$

Forecast residuals:

$$e_k(k), e(k-1), \dots, e(k-n_e)] \tag{4}$$





Seasonal lags of streamflow and residuals:

$$y(k-s), y(k-2s), \ldots, y(k-n_s \cdot s), \tag{5}$$

$$e(k-s), e(k-2s), \ldots, e(k-n_s \cdot s) \tag{6}$$

The model's general transfer function is expressed as Ec(7):

$$y(k+1) = fW \begin{bmatrix} y(k), y(k-1), \ldots, y(k-n_y); \\ u_1(k), u_1(k-1), \ldots, u_1(k-n_{u1}); u_r(k), u_r(k-1), \ldots, u_r(k-n_{ur}) \\ e_k(k), e(k-1), \ldots, e(k-n_e)] \\ y(k-s), y(k-2s), \ldots, y(k-n_s \cdot s), \\ e(k-s), e(k-2s), \ldots, e(k-n_s \cdot s) \end{bmatrix} \tag{7}$$

Where $n_y$, $n_{ur}$, $n_e$, and $n_s$ represent the autoregressive, exogenous, residual, and seasonal lag orders, respectively; $s$ denotes seasonal periodicity; and $fW$ is a nonlinear mapping function parameterized by model weights $W$.

This hybrid configuration enables simultaneous modeling of high-order dependencies, memory effects, and seasonal recurrence. Incorporating lagged residuals and seasonal components allows the model to capture patterns not accounted for by direct input–output relations, enhancing its capacity to learn both deterministic and stochastic dynamics.

An open-loop configuration was adopted during training, feeding observed streamflow values back to stabilize learning. For validation and operational forecasting, a closed-loop mode recursively propagates forecasted values. Residual sequences were

standardized during validation and testing to ensure numerical stability.

The structure of the SARIMA model was determined through visual inspection of autocorrelation (ACF) and partial autocorrelation (PACF) plots of the monthly streamflow series. The orders (p,d,q) and (P,D,Q) were selected to reflect the presence of short-term dependencies and seasonal dynamics. Differencing was not applied, as the series appeared stationary in both its level and seasonal pattern. To support this assessment, the Phillips–Perron test was applied to evaluate the presence

of unit roots and inform the choice of d=0 and D=0. A seasonal period of s=12 was defined based on the observed annual cycle in streamflow behavior.

Data were partitioned into 70% training, 15% validation, and 15% testing subsets, preserving temporal ordering to avoid information leakage; a blocked temporal split was preferred over random sampling. Alongside Bayesian regularization, early stopping was implemented by monitoring validation loss with a patience threshold of 100 epochs, halting training upon

stagnation in performance.

Model performance was evaluated with complementary metrics reflecting accuracy and parsimony. RMSE quantified average prediction error magnitude and $R^2$ assessed explained variance. For NN-SARIMAX, model order selection criteria (AIC, BIC) balanced goodness-of-fit against model complexity.




## 2.6 Evaluation of the Models

### 2.6.1 Forecast Error Degradation Across Prediction Horizons

Forecasting accuracy typically decreases as the prediction horizon extends, a well-documented phenomenon referred to as forecast error degradation (Xiang et al., 2024). To assess this behavior, the best-performing NN-SARIMAX model was evaluated over multiple lead times using RMSE and $R^2$ as performance metrics. Forecasts were produced for 2 to 24 months ahead in increments of two months, enabling the quantification of predictive accuracy across short, medium, and long term horizons.

To capture the evolution of forecast quality, RMSE and $R^2$ values were analyzed in pairs of consecutive horizons. This pairing facilitated the visualization of degradation trends through line plots, offering insights into the temporal limits of model effectiveness. The resulting performance curves contribute to a clearer understanding of how predictive uncertainty accumulates over time, which is critical for operational decision-making in hydrological forecasting.

### 2.6.2 Residual Analysis

Residual diagnostics are essential to evaluate model adequacy by verifying whether prediction errors conform to standard statistical assumptions. Residuals $e_t = y_t - \hat{y}_t$ were computed as the difference between observed and forecasted streamflow values, forming a time series subjected to rigorous statistical and graphical analysis.

Normality was tested using the Jarque–Bera test (Jarque and Bera, 1980), assessing whether residuals follow a Gaussian distribution, a prerequisite for valid inference and uncertainty quantification. Autocorrelation was examined through the Durbin–Watson statistic (Durbin and Watson, 1950), where values near 2 suggest the absence of significant serial dependence, thus confirming that temporal structures have been effectively captured. Heteroscedasticity was assessed via the Breusch–Pagan test (Breusch and Pagan, 1979); significant p-values indicated non-constant residual variance, potentially undermining the model's generalizability.

Complementing these tests, graphical diagnostics were employed to detect patterns indicative of model misspecification. Histograms with kernel density estimation (Silverman, 1986) visually assessed normality, while residuals versus fitted values plots revealed heteroscedastic patterns or structural biases. Quantile–quantile plots (Wilk and Gnanadesikan, 1968) further tested alignment between empirical and theoretical residual quantiles. Finally, ACF plots identified lagged dependencies, where the absence of significant peaks outside confidence bounds supported the white noise assumption.

## 3 RESULTS AND DISCUSSIONS

The experimental results are organized into three subsections: (1) Description of the streamflow data, (2) Determination of the pool predictors, and (3) Forecasted streamflow series. Each subsection presents and discusses relevant findings, emphasizing the identification of the most suitable forecasting model based on accuracy metrics. A comparative analysis of the four




evaluated models highlights their strengths and limitations, guiding the selection of the most effective model for streamflow
prediction.

Long-range streamflow forecasting is a valuable tool for water resource managers, enabling proactive decision-making and
strategic planning. However, excessively extending the forecast lead-time tends to degrade model performance, often resulting
in diminished predictive accuracy. Consequently, this study adopts a lead-time of 24 months to evaluate and compare
forecasting models. This horizon provides a balance between planning requirements and achievable accuracy for watershed
management stakeholders.

## 3.1 Description of the Streamflow Data

### 3.1.1 Statistical and Graphical Description

The streamflow time series at El Playón station exhibits pronounced interannual and seasonal variability, with values ranging
from 2.5 m³/s to 280 m³/s and a mean of 92.4 m³/s. This amplitude is consistent with the monomodal rainfall regime
characteristic of the region (Urrea et al., 2019). A standard deviation of 66.4 m³/s and a coefficient of variation of 71.8%
confirm the high dispersion and temporal heterogeneity of the data, which poses a challenge for accurate modeling and
prediction.

Trend detection tests were also applied. A visual inspection suggested a potential breakpoint in January 2019; however, the
Pettitt test yielded a p-value of 1.99, indicating that this change is not statistically significant. Likewise, Sen's slope and the
Mann-Kendall test indicate a slight upward trend in discharge, but without statistical significance (p = 0.3445), suggesting the
absence of a persistent monotonic trend (see Table 1 and Fig. 4).

**Table 1: Basic statistics and trend characterization for El Playón station flow rates.**

| Mean | Min | Max | Standard deviation | Coefficient of variation | Break poin | Pettitt (p-value) | Sen slope | Mann-Kendall (p-value) |
|------|-----|-----|--------------------|--------------------------|------------|-------------------|-----------|------------------------|
| 92.4 | 2.5 | 280 | 66.4 | 71.8 | Jan/2019 | 1.99 | 0.0188 | 0.3445 |



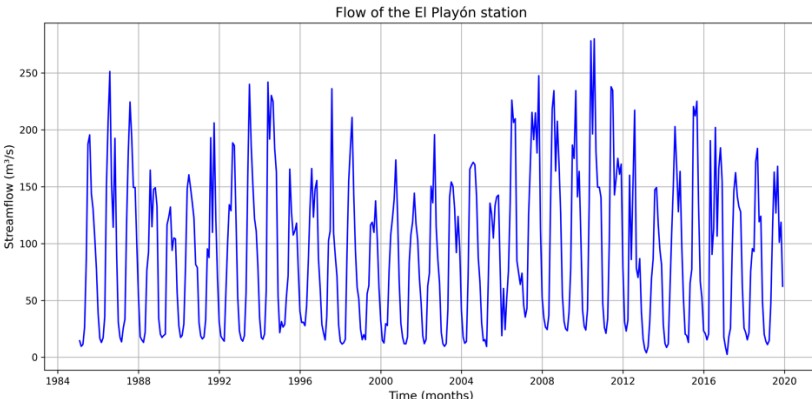

**Figure 4: Temporal behavior of streamflow at the Tocaría River.**

These characteristics highlight the complexity of the time series and justify the need for rigorous preprocessing, including stationarity and autocorrelation analysis.

### 3.1.2 Stationarity Analysis

Stationarity was assessed through both graphical decomposition and formal statistical testing. The decomposition revealed a marked seasonal component and residuals without systematic trend or variance shifts, supporting weak stationarity assumptions component lacking any systematic trend, indicating time-invariance in mean and variance (Figure 5).





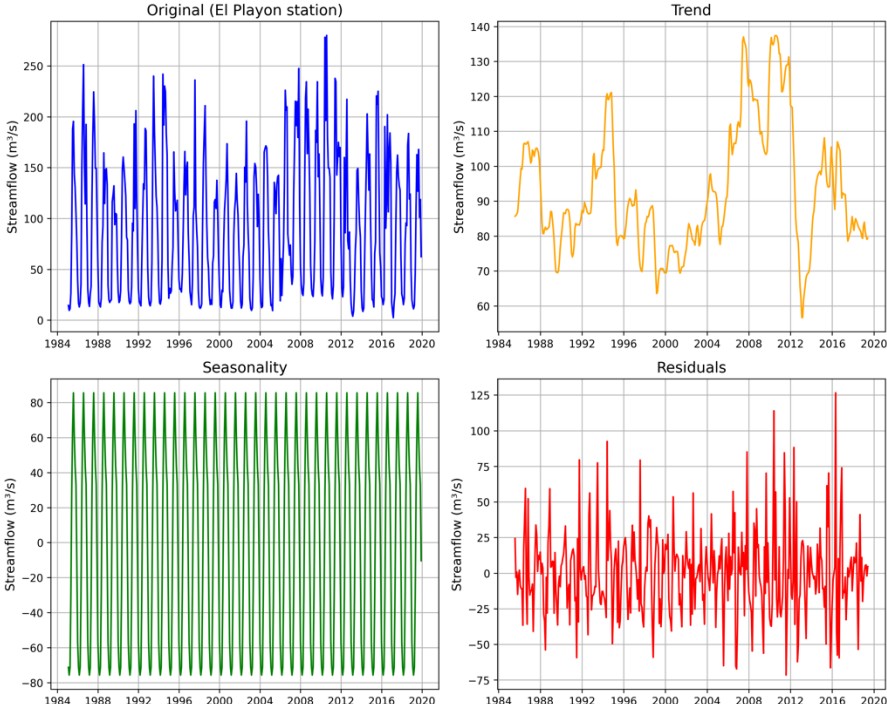

**Figure 5: Decomposition of El Playón streamflow time series into trend, seasonality, and residual components.**

The Phillips-Perron test further confirmed stationarity, yielding a test statistic of -6.5005 and a p-value < 0.0001, allowing rejection of the null hypothesis of a unit root. These results justify modeling without differencing and the selection of d = 0 in the SARIMA model.

**3.1.3 Autocorrelation and Partial Autocorrelation Structure**

The ACF displays significant periodic peaks at regular intervals, indicative of strong seasonal periodicity and persistent temporal dependencies beyond short lags (Fig. 6). This cyclicality aligns with hydrological patterns driven by seasonal precipitation and runoff dynamics, underscoring the appropriateness of seasonal forecasting models such as SARIMA.





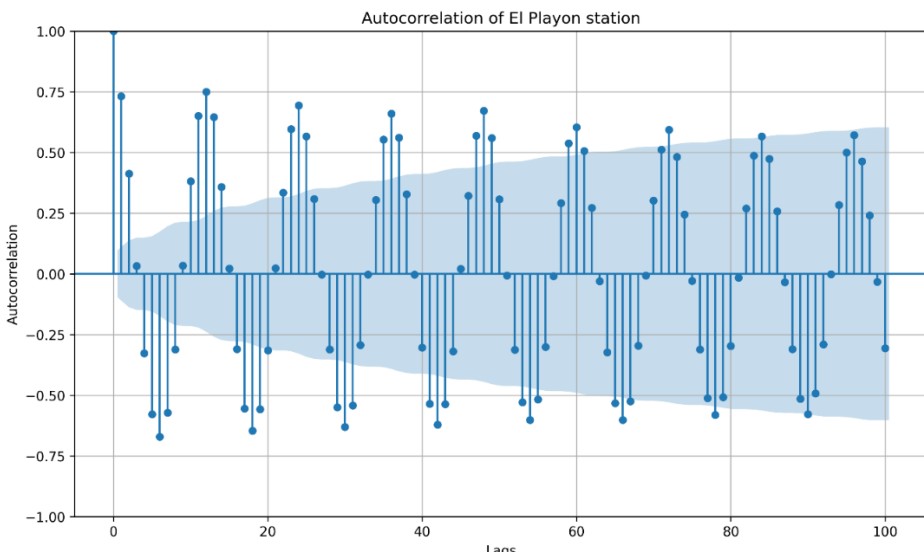

**Figure 6: Autocorrelation of the time series of El Playón streamflow for 100 months of lags.**

The PACF shows a prominent spike at lag 1 followed by rapid decay to values within the confidence bounds (Fig. 7), consistent with an autoregressive process of order 1 (AR(1)) though the selected SARIMA model incorporates higher-order dynamics to capture the longer memory and seasonal structure identified in the ACF. The lack of significant PACF values at higher lags indicates limited benefit from additional autoregressive terms.

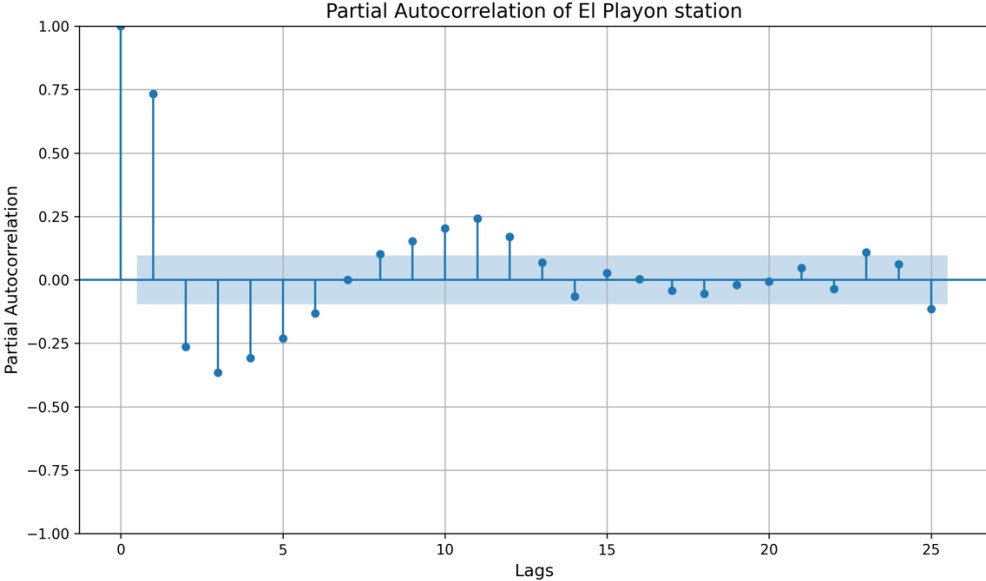

**Figure 7: Partial autocorrelation function of El Playón streamflow series over 25 monthly lags.**




Based on combined ACF and PACF analysis, a SARIMA $(4,0,4)(0,0,3)_{12}$ model was selected. This model configuration incorporates the identified seasonal periodicity (season length $s = 12$), confirms stationarity ($d = 0$), and captures short-term autoregressive and moving average dynamics. Residual diagnostics validated the model's adequacy, demonstrating no significant autocorrelation.

## 3.2. Determination of the Pool Predictors

### 3.2.1 Selection of precipitation NLPCs as exogenous variables

The 81 CHIRPS precipitation time series for the Tocaría River basin were reduced to two nonlinear principal components (NLPC1 and NLPC2) using NLPCA. NLPC1 explains 92.5% of the total variance (eigenvalue = 25.6), and NLPC2 accounts for 7.4% (eigenvalue = 2.1), preserving the dominant spatiotemporal patterns of the original dataset retaining over 99% of the total variance and the key spatiotemporal signatures of basin-wide precipitation.

Spearman correlation analysis revealed moderate positive correlations between the NLPCs and the streamflow observed at the El Playón station, with coefficients of 0.50 for NLPC1 and 0.34 for NLPC2. These results support the predictive utility of the NLPCs as exogenous variables, consistent with findings by Ocampo-Marulanda et al. (2025), who used CHIRPS-derived precipitation components to reconstruct streamflow time series in the Cravo Sur basin under data-scarce conditions. These components were subsequently included as covariates in the streamflow forecasting model, enabling a more parsimonious yet informative representation of the basin's precipitation dynamics

Incorporating NLPCs as exogenous predictors offers a compact yet informative representation of precipitation, reducing dimensionality without sacrificing critical hydrometeorological signals.

### 3.2.2 Selection of Macroclimatic Variables as Exogenous Variables

The influence of 21 MVs on the streamflow variability of the Tocaría River was assessed using a lagged Spearman correlation analysis, considering time lags from 0 to 14 months. This asynchronous framework allowed for identification of delayed relationships between climate anomalies and hydrological responses, which are crucial for predictive modeling. This lagged correlation framework enables detection of delayed hydroclimatic teleconnections, which are essential for long-lead streamflow forecasting.

Statistical significance was evaluated using two-tailed tests with confidence levels between 90% and 99%. Figure 8 displays the lagged correlations between the MVs and the streamflow at the El Playón station. Several variables demonstrated statistically significant correlations across multiple lags, exhibiting a quasi-periodic structure that reflects the hydroclimatic seasonality modulated by the Intertropical Convergence Zo ne (ITCZ).



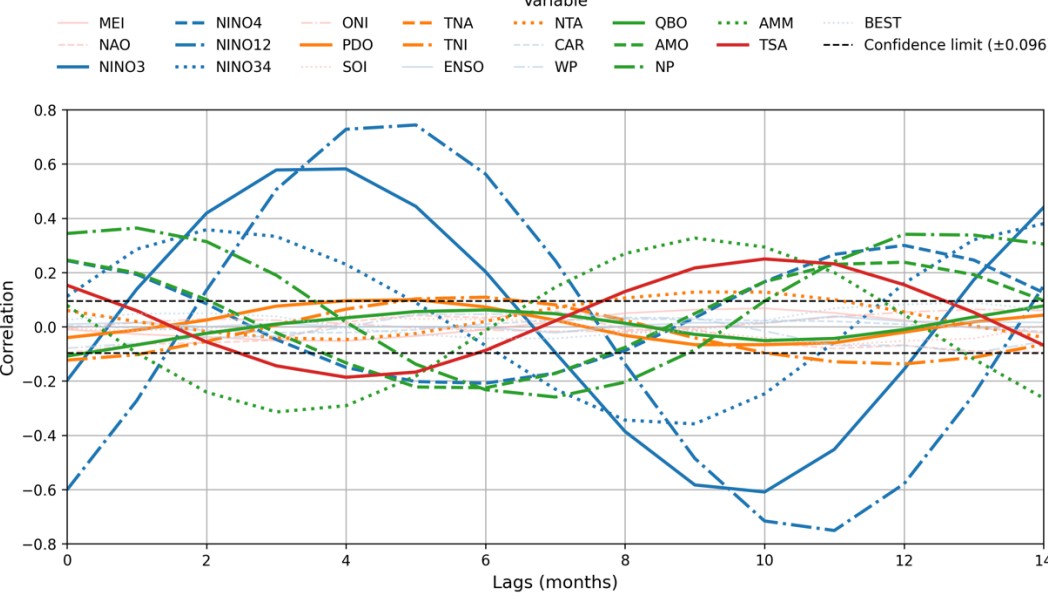


**Figure 8: Lagged Spearman correlations between MVs and streamflow at El Playón station.**

Key Pacific-related indices—namely NINO12 (ρ = 0.75) and NINO3 (ρ = 0.60)—emerged as the strongest predictors, underscoring the robust influence of ENSO-related sea surface temperature (SST) anomalies on the Tocaría River streamflow.

These findings are consistent with previous studies across Colombia (e.g., Canchala, 2020c; Cadavid and Salazar, 2008; Poveda et al., 2002), which reported comparable lagged correlations between ENSO indices and regional streamflow, often with cyclic behavior.

In addition, Atlantic-related variables such as AMM (ρ = 0.32) and TSA (ρ = 0.25) showed statistically significant, albeit weaker, correlations. These results corroborate the relevance of Atlantic SST variability in eastern Colombia. The Llanos low-

level jet and moisture transport from the tropical Atlantic—modulated by indices such as AMM and TSA—have been highlighted as key controls of precipitation and streamflow in the Orinoquía region (Labat et al., 2012; Nieto et al., 2008; Builes et al., 2022; Correa et al., 2024).

The set of significantly correlated predictors spans both Pacific and Atlantic basins: NINO3, NINO4, NINO12, NINO34, PDO, TNI, NP, QBO (Pacific-related), and AMO, NTA, TNA, NAO, AMM, TSA (Atlantic-related). This reinforces the complexity

of the climate–hydrology interactions in the Tocaría River basin and emphasizes the need to account for both direct and lagged macroclimatic signals in predictive modeling.

### 3.2.3 Assessing Multicollinearity and Variable Selection for Improved Time Series Forecasting

To ensure model robustness and avoid overfitting, multicollinearity among predictors was quantified using the VIF. Table 2 presents the VIF values for all candidate variables. While most variables showed acceptable levels (VIF < 10), certain



predictors; especially those related to ENSO; exhibited extreme multicollinearity. NINO3, NINO4, NINO12, and NINO34 had VIFs ranging from 2,492 to 81,825. TNA and TSA presented infinite VIFs, indicating perfect collinearity with other variables most likely due to structural redundancy among Atlantic SST indices, as confirmed by the correlation heatmap.

**Table 2: Variance Inflation Factor Analysis for predictor variables.**

| Variable | VIF | Variable | VIF |
|---|---|---|---|
| NAO | 2 | NTA | 16 |
| NINO3 | 38,899 | QBO | 1 |
| NINO4 | 29,126 | AMO | 8 |
| NINO12 | 2,492 | NP | 17,705 |
| NINO34 | 81,825 | AMM | 8 |
| PDO | 2 | TSA | inf |
| TNA | inf | PNLPCA_PC1 | 4 |
| TNI | 4 | PNLPCA_PC2 | 4 |


These results, along with the correlation heatmap (Fig. 9), confirm substantial interdependence among ENSO indices, suggesting that including multiple related predictors may lead to instability and redundancy. For example, NINO3, NINO4, and NINO34 are highly correlated, and thus only one may be sufficient to represent ENSO dynamics. Likewise, high correlation between TNA and TNI implies redundancy.








**Figure 9: Correlation heatmap of predictor variables.**

To address this, various combinations of variables were tested through an iterative model refinement process, prioritizing
predictors with strong individual relationships to streamflow and minimal mutual collinearity. This selection strategy balances
predictive power with model parsimony, mitigating multicollinearity-induced variance inflation and enhancing
generalizability. Model performance was evaluated using RMSE and $R^2$ to balance predictive accuracy and parsimony.

Ultimately, the optimal subset of exogenous predictors was determined to be:
PNLPCA_PC1, PNLPCA_PC2, NINO4, NINO12, NP, AMO, TNA, AMM, and TSA.

This final configuration minimized multicollinearity, preserved critical information from both precipitation and climate
variability, and improved the interpretability and reliability of the streamflow forecasting model.



The final pool of predictors; comprising key macroclimatic indices and the first two NLPCs; provided the input for the forecasting models evaluated in the subsequent section. The next subsection presents the performance comparison of candidate models and the resulting streamflow forecasts.

## 3.3 Forecasted Streamflow Series

The forecasting performance of the four models was evaluated using $R^2$, RMSE, AIC, and BIC. All models were trained and validated on independent datasets to ensure generalization. Table 3 summarizes the results. All configurations yielded adequate performance in forecasting monthly streamflow up to 24 months in advance. As expected, $R^2$ values decreased and RMSE increased from training to validation. Training $R^2$ ranged from 0.85 to 0.91, while validation $R^2$ dropped to 0.71–0.78. RMSE increased from 26.8–32.7 in training to 41.5–49.9 in validation.

The model combining ocean-atmospheric indices with nonlinear components showed the smallest degradation between training and validation, indicating higher robustness. In contrast, the baseline and the model using only nonlinear components yielded lower predictive skill, though both maintained acceptable performance (validation $R^2$ of 0.72 and 0.71, respectively), which is notable given the hydroclimatic variability of the basin.

The best results were obtained with the combined model, which achieved the highest $R^2$ (0.91 training, 0.78 validation) and the lowest RMSE (26.8 and 41.5). No other model exceeded 0.87 in training or 0.73 in validation. All other models reported training RMSE above 29.0 and validation RMSE above 44.4. The combined configuration consistently showed lower errors.

In terms of model complexity, AIC was identical (3103) for the three simpler configurations, while slightly higher (3110) in the combined model. BIC increased more noticeably in the combined case (3173), reflecting its additional structural complexity.

Although the inclusion of nonlinear components alone did not significantly enhance the baseline, their integration with ocean-atmospheric variables resulted in a marked improvement in accuracy and consistency across datasets.

**Table 3: Forecast performance metrics for the four streamflow forecasting models.**

| Model | Exogenous variables (inputs) | Training | | Validation | | AIC | BIC |
|---|---|---|---|---|---|---|---|
| | | $R^2$ | RMSE | $R^2$ | RMSE | | |
| SARIMA $(4,0,4)(0,0,3)_{12}$ | None | 0.86 | 30.2 | 0.72 | 49.9 | 3103 | 3122 |
| NNSARIMAX-MV | MV | 0.87 | 29.0 | 0.73 | 44.4 | 3103 | 3129 |
| NNSARIMAX-NLPC | NLPC | 0.85 | 32.7 | 0.71 | 49.7 | 3103 | 3129 |
| NNSARIMAX-MVNL | MV and NLPC | 0.91 | 26.8 | 0.78 | 41.5 | 3110 | 3173 |



These results are consistent with previous findings, Table 4, in the literature. Reported R² values for monthly streamflow prediction range from moderate (~0.65) to near-perfect (~0.99), depending on lead time, predictor variables, and catchment complexity.

For instance, Hosseinzadeh et al. (2023) reported R² = 0.92 using a multivariate SARIMAX model with temperature and precipitation as exogenous inputs in the Colorado River Basin. Cheng et al. (2020) achieved R² = 0.95 using an LSTM model, although this performance was limited to one-month-ahead forecasts and dropped significantly (R² < 0.20) at longer lead times. Several hybrid or AI-based approaches have yielded high predictive accuracy (R² ≈ 0.99), such as those reported by Fathian (2019), Moeeni et al. (2017), and Dariane (2024). However, these results often stem from short-term horizons, multi-source calibration, or highly tuned model configurations prone to overfitting. Ghorbani (2016) obtained R² between 0.77 and 0.84 using ANN and SVM models, while Gómez (2010) reported a more moderate R² ≈ 0.64 for the Bogotá River basin.

In comparison, the performance of our models; particularly the NN-SARIMAX-MVNL; stands competitively within this broader context. Despite relying on a single hydrometric station and forecasting beyond the one-month horizon, our approach demonstrates robust skill in capturing the high temporal variability and structural complexity of streamflow in the Tocaría River Basin.

**Table 4: Validation R² of selected monthly streamflow forecasting models from literature.**

| Best model | Description | Validation R² | References |
|---|---|---|---|
| SARIMAX | 24-month forecasts with temperature and precipitation | 0.92 | Hosseinzadeh, et al., 2023 |
| LSTM | One-month-ahead forecasts | 0.95 | Cheng et al., 2020 |
| MARS1-SETAR | Hybrid nonlinear time series and AI modeling | 0.99 | Fathian, et al 2019 |
| SARIMA-ANFIS | SARIMA combined with neuro-fuzzy and ANN systems | 0.94 | Moeeni, 2017 |
| Multistep data-driven | Neural-fuzzy hybrid with input selection | 0.97 | Dariane, 2017 |
| Hybrid model | SARIMA combined with nonlinear ANN | 0.72 | Moeeni, 2016 |
| ELM | Extreme Learning Machine approach | 0.82 | Yassen et al., 2016 |
| RBF | MLP, RBF, and SVM models for river flow prediction | 0.87 | Ghorbani, 2016 |
| PMC 20 NCO | Neuro-fuzzy vs neural networks for Bogotá River | 0.65 | Gomez, 2012 |





### 3.3.1 Statistical Evaluation of the Best Model for Streamflow Forecasting

Figure 10 shows the streamflow prediction results for the El Playón station using the NN-SARIMAX-MVNL model, which
integrates exogenous variables including MV's and NLPCs derived from local precipitation data. The figure includes three
key elements: observed streamflow, model predictions for the training period, forecasts for the validation period, and test
period.

The observed streamflow series reveals pronounced seasonality, characterized by recurrent peaks and troughs consistent with
the hydroclimatic regime of the basin. During the training phase, the model reproduces the seasonal dynamics and intra-annual
variability with high accuracy, showing minimal deviations. In the validation period, although some differences appear;
particularly underestimations of extreme highs or delays in response; the general flow patterns, including the timing and
intensity of seasonal peaks, are well preserved. These results highlight the model's capacity to capture both historical and
prospective streamflow behaviors based on relevant exogenous predictors.

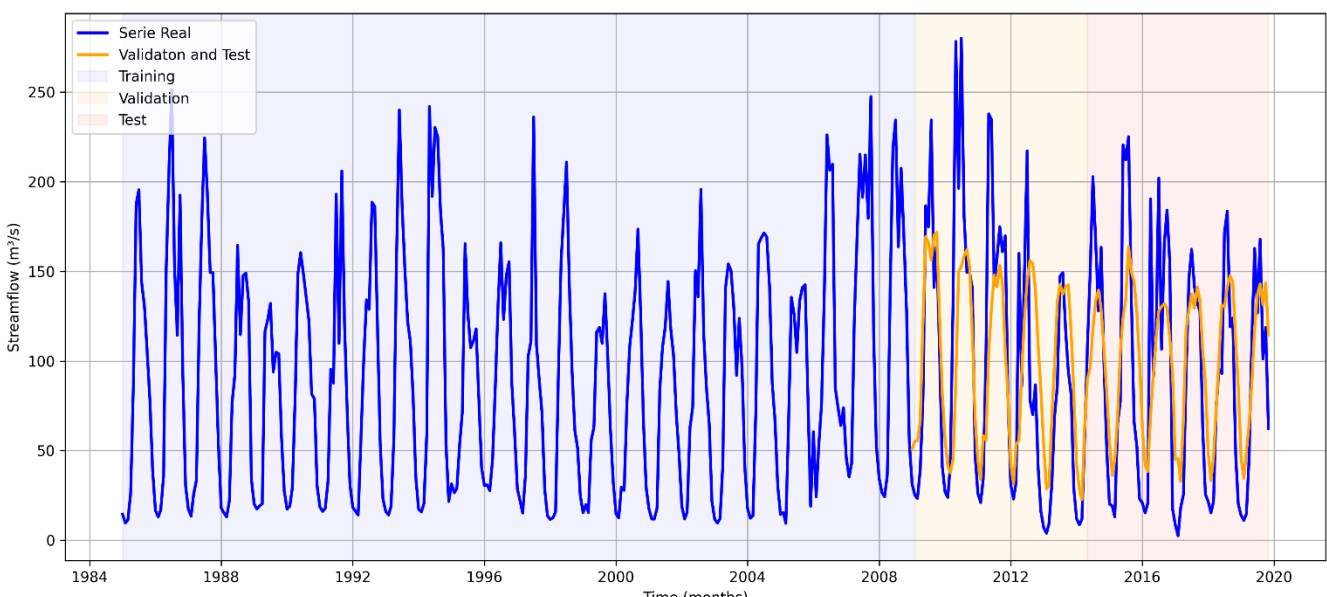


**Figure 10: Observed and predicted streamflow for training and validation using the NN-SARIMAX-MVNL model.**

To evaluate the model's temporal generalization capability, streamflow forecasts were generated for horizons up to 24 months
(Fig. 11). The RMSE remained below 11 m³/s for the first four months, which is low relative to the standard deviation of the
observed series (σ ≈ 66 m³/s). Between months 5 and 8, RMSE increased to approximately 30–50 m³/s and then stabilized.
The R² metric remained above 0.90 up to month 7, then gradually declined, reaching 0.83 by month 10 and remaining above
0.75 throughout the entire 24-month forecast horizon.

These results confirm the robustness of the NN-SARIMAX-MVNL model for medium- and long-term forecasting, maintaining
a favorable balance between accuracy and lead time. The performance is suitable for practical decision-making applications
in water resource planning, including drought early warning and integrated watershed management in the Tocaría River basin.

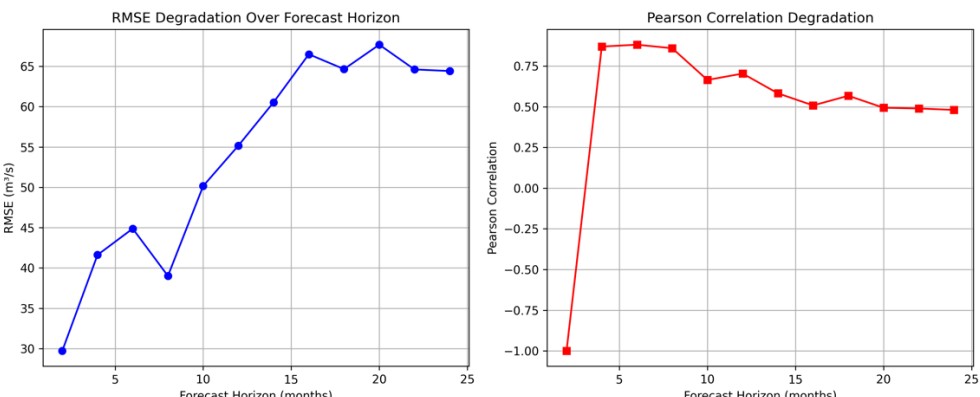

**Figure 11: Forecast error degradation across prediction horizons based on RMSE and R² metrics.**


A residual analysis was performed to assess the adequacy of the model and detect any remaining structure in the errors.
Residuals were computed as the difference between observed and predicted streamflows. The residual series is generally
centered around zero, with sporadic peaks reaching +150 m³/s and troughs of −100 m³/s, primarily associated with abrupt
hydrological events.

Several statistical diagnostics were applied to evaluate the residual distribution. The Jarque–Bera test yielded a statistic of 13.2
(p = 0.0014), suggesting a moderate deviation from normality. The Q–Q plot reveals mild departures from the theoretical
normal line, particularly in the upper quantiles, indicating that extreme residuals are somewhat asymmetrically distributed
(Fig. 12). Overall, while the residuals are not perfectly normal, their distribution remains reasonably close to symmetry for
most values.

The Durbin–Watson statistic of 1.72 indicates an absence of significant autocorrelation, which is corroborated by the ACF
plot, where no substantial lags exceed the confidence bounds. This implies that the model has adequately captured the temporal
dependencies in the streamflow series, and that residuals are largely uncorrelated.

The Breusch–Pagan test returned a statistic of 19.05 (p = 0.0396), suggesting the presence of heteroscedasticity. This is further
supported by the residuals vs. fitted values plot, where residual variance appears non-constant, particularly for low to mid-
range predicted values. Clustering and trends in residuals are visible in the initial portion of the fitted value range, consistent
with heteroscedastic behavior. These findings indicate that while the model performs well overall, its variance may vary across
the prediction space, particularly during high variability periods.



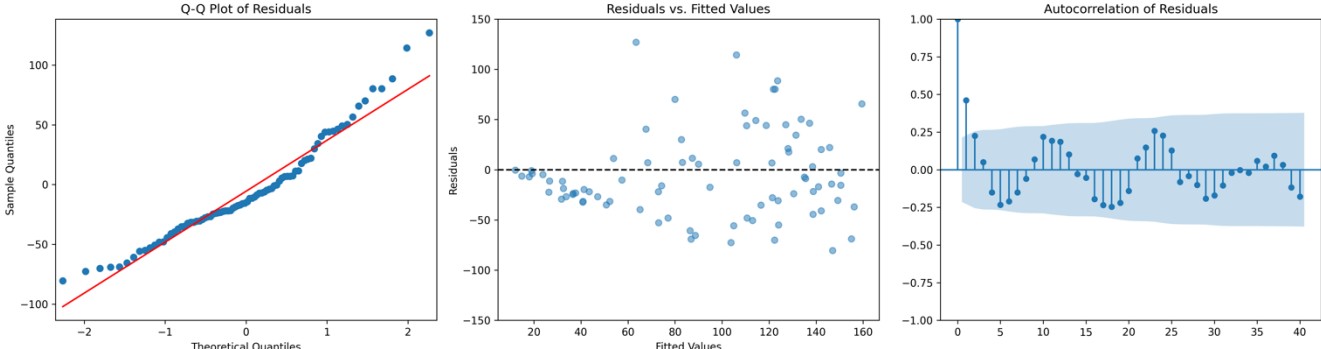


**Figure 12: Graphical analysis of normality (Q-Q-Plot of residuals), heteroscedasticity (scatter plot, Residuals vs Fitted values), and ACF of residuals.**

In summary, the statistical analysis supports the conclusion that the NN-SARIMAX-MVNL model provides reliable
streamflow forecasts under average and low-flow conditions. However, its ability to accurately predict high-magnitude peak flows remains limited, likely due to complex hydrometeorological drivers not fully captured by the selected predictors. Despite this, the model demonstrates strong potential for operational use in drought risk assessment and water resource management, particularly for forecasting low-flow events in the Tocaría River basin.

## 4 CONCLUSIONS

This study conducted a rigorous comparative assessment of four monthly streamflow forecasting models for a data-scarce basin, all based on the SARIMA framework. These models were incrementally enhanced through the integration of exogenous predictors and ANNs to improve predictive skill. All configurations demonstrated satisfactory performance, as reflected in high $R^2$ and low RMSE across both calibration and validation phases. However, the baseline SARIMA $(4,0,4)(0,0,3)_{12}$ model, while capable of capturing seasonal and autoregressive patterns, showed limited forecasting ability when compared to models
incorporating external predictors.

Among the configurations tested, the SARIMAX models with MVs, and the combination of MVs with NLPCs; substantially outperformed the baseline. The NN-SARIMAX-MVNL model, which merges both large-scale climate signals and localized precipitation variability, achieved the highest forecasting accuracy and stability, consistently yielding the lowest RMSE and highest $R^2$ in both training and out-of-sample testing.

The selection of exogenous predictors was grounded in hydroclimatic relevance. The MVs included indices such as NINO4, NINO12, NP, AMO, TNA, AMM, and TSA, which collectively capture dominant modes of Pacific and Atlantic ocean–atmosphere variability affecting the Tocaría River basin. At the local scale, precipitation data derived from the CHIRPS dataset



were incorporated via NLPCA, allowing the model to account for spatially distributed and nonlinear hydrometeorological influences.

The NN-SARIMAX-MVNL model demonstrated strong predictive performance at extended lead times of up to 24 months, highlighting its applicability for anticipatory water resource management and hydroclimatic risk mitigation in data-limited settings. Residual diagnostics confirmed its reliability in forecasting both low; and average-flow regimes; an essential feature for drought preparedness and long-term planning in the basin.

In summary, this work underscores the value of integrating large-scale climate teleconnections with locally resolved

precipitation dynamics to enhance streamflow predictability under conditions of data scarcity. The hybrid modeling framework proposed herein offers a scalable and transferable approach for streamflow forecasting in other poorly instrumented basins, with implications for strengthening adaptive watershed governance in the context of climate variability and change.

## 5 AUTHOR CONTRIBUTION

Conceptualization: JDSO, COM, LMCA, TC, and TAFE; data curation: JDSO and COM; formal analysis: JDSO, COM;
LMCA, TC, and TAFE; Funding acquisition: LMCA; Methodology: COM, TC, and TAFE; Project administration: LMCA; Resources: LMCA and TAFE; Software: JDSO, COM; Supervision: LMCA, TC and TAFE; Validation: TC and TAFE; Visualization: JDSO and COM; Writing (original draft preparation): JDSO, COM, LMCA and TC; and Writing (review and editing): JDSO, COM; LMCA, TC, and TAFE.

## 6 COMPETING INTERESTS

The authors also have no other competing interests to declare.

## 7 ACKNOWLEDGEMENTS

The authors would like to thank to the Research Group TERRANARE of the Fundación Universitaria de San Gil, the Interdisciplinary Forecasting Research Oriented Group - IFROG of the Federal Rural University of Pernambuco and the Environmental Engineering Group - GIA of the Universidad Mariana for their contributions to this research work. Finally,
thanks to IDEAM, the United States Geological Survey USGS, and the University of California, Santa Barbara (UCSB) for providing the database containing flow data for the Colombian Orinoquía and precipitation data from CHIRPS.

## 8 FINANCIAL SUPPORT

The project "Fortalecimiento de los sistemas de información de la calidad y valoración de los servicios ambientales del agua para contribuir al desarrollo sostenible del sector agroindustrial del departamento de Casanare. Yopal – Nunchía" (BPIN Code:



2020000100435) was funded with resources from the Ministerio de Ciencia, Tecnología e Innovación de Colombia – CTeI, through the fund of the Sistema General de Regalías – SGR.

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
