# Peer review of "Use of nonlinear principal components of CHIRPS precipitation data and ocean-atmospheric variables for streamflow forecasting in an area of scarce data. Case study, Tocaría river basin - Orinoquia Colombiana"

_EGUsphere, 2025_

## Referee Comment (RC2)

**Dear Authors,**

I have carefully read your manuscript "Use of nonlinear principal components of CHIRPS precipitation data and ocean-atmospheric variables for streamflow forecasting in an area of scarce data".

Let me begin by saying that the topic is very relevant, especially considering the challenges of forecasting hydrology in data-scarce tropical basins. Your motivation is solid, the literature review is rich, and the scientific objectives are meaningful.

However, the manuscript requires substantial revisions before it can be considered for publication. My comments aim to be constructive, clear, and helpful. I divide them into major issues and more specific comments.

**Issues**

The description of the hybrid model is not sufficiently precise for reproducibility. It is unclear:

- How exactly the ANN interacts with SARIMA (residual modelling? parallel modelling? combined loss?).
- What is the ANN architecture (layers, activation functions, training epochs, optimizer).
- How exogenous variables are lagged to avoid information leakage.
- How the model moves from open-loop to closed-loop.

A fully explicit mathematical formulation and a schematic model diagram are required.

The autoencoder used for nonlinear PCA is described only conceptually. Please specify:

- Number of hidden layers and units
- Type of activation functions
- Loss function
- Training epochs
- Optimizer
- Reconstruction error achieved
- Rationale for selecting exactly two nonlinear components

At the moment, NLPCA cannot be reproduced from the information provided.

You report remarkably high R2 values (0.75–0.78) for 24-month-ahead forecasts.

This is unusual for tropical hydrology, where long-lead predictions are extremely difficult.

**Please:**

- Include a baseline benchmark vs climatology and persistence
- Provide an alternative validation in which the training/validation split is shifted forward in time
- Discuss the potential risk of overfitting when many exogenous predictors are used for a single station
- Verify that no future information enters through the NLPCA step

Table 2 shows extreme VIF values reaching 81,825, and two variables with infinite VIF.

However, the process of variable selection is narrated but not clearly documented.

**I suggest:**

- Provide a clear step-by-step table describing which variables were removed and why
- Avoid keeping multiple ENSO indices that are functionally redundant
- Confirm that the final chosen set has acceptable VIF values

This will make your variable selection reproducible.

Pettitt p-value is reported as 1.99, which is impossible (p-values must be  $\leq 1$ ).

Please revise all statistical outputs and their interpretations.

The Breusch–Pagan test indicates heteroscedasticity, and the residuals vs. fitted plot confirms this. Please discuss or explore:

- Log transformation of discharge
- Box–Cox transformation
- Weighted regression or heteroscedasticity-aware models
- Impact on long-term forecast uncertainty

Forecasts are presented only as point estimates.

**Please include:**

- Confidence intervals
- Prediction intervals
- Bootstrapped ensembles
- At minimum, a written discussion about uncertainty

This is essential in hydrological forecasting.

**NLPC1 and NLPC2 remain abstract.**

**Please include:**

- Spatial loading maps
- Seasonal cycle of each component
- Their relationship to ENSO/ITCZ migration
- Interpretation of their hydrological meaning

This would strengthen the scientific contribution.

**Figure 1**

- Caption should be more descriptive.
- Ensure all acronyms in the image are defined.

**Figure 2**

• In IS the units for kilometers are km not Kms

**Table 1**

- As said, Pettitt p-value impossible (1.99).
- Units missing for mean, min, max, and standard deviation (m3/s).
- The title should specify "Streamflow characteristics at El Playón station".

**Table 2 (VIF)**

Should include a final column indicating whether each variable was retained or dropped.

**Table 3**

- Units missing for RMSE.
- AIC/BIC interpretations should be provided in a footnote.

**Table 4**

• Clarify if these R2 values are from training, validation, or test datasets.

**Stationarity section needs rewriting**

Some expressions are grammatically incorrect:

"supporting weak stationarity assumptions component lacking any systematic trend" Please rewrite this entire subsection more clearly.

**Improve logic flow between sections**

**Sometimes:**

- The same idea is repeated (e.g., ENSO relevance).
- Paragraphs begin with generic phrases ("These results confirm...").

A more synthetic writing style would improve clarity.

**Please check all units**

I found several missing or unclear units:

**I strongly recommend adding:**

- A short description of software used (Python, R, MATLAB).
- Version numbers for key libraries (TensorFlow, PyTorch, statsmodels, etc.).

• Pseudocode or model-training flowchart.

My recommendation is Major Revision.

I hope my comments help you strengthen your manuscript. I would be very happy to re-evaluate a revised version.

---

## Author Comment (AC1)

**COMMENTS FOR THE AUTHOR:**

**Response to Reviewer 1**

We would like to thank the reviewer for his/her helpful comments. Thank you. All of your comments have been taken into consideration, and the paper was modified accordingly. Please find below our responses.

Comment 1: Section 2.5:

- It would be beneficial to provide a detailed explanation of the Neural Network parameters and the architecture of the NN-SARIMAX model to help readers easily and clearly understand the structure of the neural network.

Answer: A detailed description of the neural network architecture and its associated parameters was added to Section 2.4.1. This description clarifies the structure of the neural network employed within the NLPCA framework, including its role in processing CHIRPS precipitation data and its integration into the SARIMAX modeling approach as an exogenous variable, thereby enhancing model transparency and reproducibility.

Comment 2: Section 3.2.1

- It is stated that two non-linear principal components (explaining 92.5% and 7.4% of the variance, respectively) were selected out of a total of 81 components, collectively explaining 99.9% of the variance. However, this approach may lead to overfitting, as it effectively considers nearly the entire variation unless the model is validated through cross-validation or other model selection criteria (e.g., AIC, BIC, etc.) to determine the optimal number of components. Therefore, it should be clearly explained how potential overfitting was assessed and mitigated.

Answer: Potential overfitting associated with the selection of NLPCs was explicitly assessed and mitigated by implementing a data-splitting strategy within the NLPCA framework. The dataset was divided into independent training, validation, and testing subsets.

- It is also unclear why only the first two principal components account for 99.9% of the variance, while the remaining 79 components contribute only 0.1%. This large discrepancy warrants further clarification.

Answer: The apparent concentration of variance in the first two nonlinear principal components arises from the use of a nonlinear PCA (NLPCA) approach implemented through an autoencoder-based neural network. Unlike linear PCA, NLPCA does not decompose variance through orthogonal eigenvectors but instead learns a low-dimensional nonlinear manifold that captures the dominant structure of the data.

Although the input consists of 81 CHIRPS grid cells, these variables exhibit strong spatial coherence. As a result, the majority of the precipitation variability can be effectively represented by two latent nonlinear components (NLPC1 and NLPC2), which together explain 100% of the reconstructed variance, while the remaining components account for negligible residual variability.

Comment 3: Section 2.4.3:

- The approach used to address multicollinearity is generally sound and viable. However, there is no clear evidence indicating that the multicollinearity issue has been fully resolved, such as through recalculating the Variance Inflation Factor (VIF) after the iterative removal of collinear and less important predictors. Many of the retained variables still exhibit extremely high VIF values (e.g., NINO4 = 29,126; NINO12 = 2,492; NP = 17,705; TNA = ∞; and TSA = ∞). It remains unclear whether multicollinearity persists among these nine predictors or not.

Answer: The assessment of multicollinearity was strengthened by applying a more stringent variable-selection procedure. Following the iterative removal of collinear and less influential predictors, the VIF was recalculated for the final set of retained meteorological variables. This additional step confirmed that multicollinearity was effectively reduced. See in the section 3.2.3.

We thank the reviewer for the thorough evaluation, constructive comments, and helpful recommendations. We have carefully addressed all observations and hope that the revisions adequately strengthen the manuscript. We would be pleased to have the opportunity for the revised version to be re-evaluated.